# Prevalence of a BRCA2 Pathogenic Variant in Hereditary-Breast-and-Ovarian-Cancer-Syndrome Families with Increased Risk of Pancreatic Cancer in a Restricted Italian Area

**DOI:** 10.3390/cancers15072132

**Published:** 2023-04-03

**Authors:** Valentina Zampiga, Ilaria Cangini, Erika Bandini, Irene Azzali, Mila Ravegnani, Alessandra Ravaioli, Silvia Mancini, Michela Tebaldi, Gianluca Tedaldi, Francesca Pirini, Luigi Veneroni, Giovanni Luca Frassineti, Fabio Falcini, Rita Danesi, Daniele Calistri, Valentina Arcangeli

**Affiliations:** 1Biosciences Laboratory, IRCCS Istituto Romagnolo per lo Studio dei Tumori (IRST) “Dino Amadori”, 47014 Meldola, Italy; 2Unit of Biostatistics and Clinical Trials, IRCCS Istituto Romagnolo per lo Studio dei Tumori (IRST) “Dino Amadori”, 47014 Meldola, Italy; 3Romagna Cancer Registry, IRCCS Istituto Romagnolo per lo Studio dei Tumori (IRST) “Dino Amadori”, 47014 Meldola, Italy; 4Surgical Department, Infermi Hospital, 47923 Rimini, Italy; 5Department of Medical Oncology, IRCCS Istituto Romagnolo per lo Studio dei Tumori (IRST) “Dino Amadori”, 47014 Meldola, Italy

**Keywords:** *BRCA*, pathogenic variant, cancer, HBOC, pancreatic cancer, NGS

## Abstract

**Simple Summary:**

Hereditary pathogenic/likely-pathogenic variants (PVs/LPVs) of *BRCA1* and *BRCA2* genes are the principal genetic cause of breast cancer (BC), ovarian cancer (OC), and other malignancies such as prostate (PrC) and pancreas (PC) carcinomas. The proportion of *BRCA1* vs. *BRCA2* is specific to various populations in different regions, and several PVs have been observed to be founders, besides recurring in narrow geographical areas. In our study, in a selected cohort of subjects characterized by a common local origin and a cancer family history, we identified a *BRCA2* PV that was further analyzed and correlated to the risk of PC onset.

**Abstract:**

PVs and LPVs in *BRCA1/2* genes are correlated to a high risk of developing breast cancer and/or ovarian cancer (Hereditary Breast and Ovarian Cancer syndrome, HBOC); additionally, in recent years, an increasing number of *BRCA 1/2* variants have been identified and associated with pancreatic cancer. Epidemiologic studies have highlighted that inherited factors are involved in 10% to 20% of PCs, mainly through deleterious variants of *BRCA2***.** The frequency of *BRCA1/2* germline alterations fluctuates quite a lot among different ethnic groups, and the estimated rate of PVs/LPVs variants in Italian HBOC families is not very accurate, according to different reports. The aim of our study is to describe the prevalence of a *BRCA2* PV observed in a selected cohort of HBOC patients and their relatives, whose common origin is the eastern coast of Emilia Romagna, a region of Italy. This study provides insight into the frequency of the variant detected in this area and provides evidence of an increased risk of pancreatic and breast cancer, useful for genetic counseling and surveillance programs.

## 1. Introduction

*BRCA1* and *BRCA2* are tumor suppressor genes involved in DNA repair and the maintenance of genomic stability. PVs and LPVs in *BRCA1/2* genes cover a percentage ranging from 20 to 25% of all hereditary breast cancers (BCs) and about 5–10% of all BCs [1] and are correlated to an increased risk of developing some types of aggressive carcinomas. Strict cancer surveillance and risk-decreasing prophylactic surgeries have become a central component in the management of *BRCA1/2* PV and LPV carriers [2]. *BRCA 1/2* PVs and LPVs are commonly associated with breast and ovarian cancers along with other malignancies involving the prostate, pancreas and stomach [3]. In particular, Pancreatic Ductal Adenocarcinoma (PDAC) represents one of the most aggressive tumors, with a 5-year survival rate of 8% [4]. Epidemiologic studies highlighted that 10% to 20% of PDACs are associated with inherited factors and with mainly deleterious variants in *BRCA1*, *BRCA2*, *PALB2* and *CDKN2A* genes [5]. Several hereditary syndromes have been associated with familial pancreatic cancer (FPC) onset, including HBOC, and the presence of deleterious variants has been reported to have therapeutic implications for patients and their families [6]. Furthermore, patients and their affected relatives can also benefit from a comprehensive and longitudinal screening program for early detection, which aims to diagnose and treat precursor lesions, including pancreatic intraepithelial neoplasia, IPMN and mucinous cystic neoplasm [7]. Recently, germline *BRCA1/2* pathogenic variants (*gBRCA1-2* PVs) have been observed to be associated with an elevated risk of developing PDAC. The prevalence of loss-of-function *gBRCA1-2* PV ranges from 5% in unselected PDAC case series to 15–20% in FPC. Reports about the epidemiology of *BRCA* PVs and LPs in Italy are present but are not so extended and massive, since data about their frequency and distribution in a more considerable Italian population, independent of age, stage, family history or any other information, are quite weak and insufficient [8]. In addition to a hereditary component, there are several risk factors for a predisposition to PDAC onset, such as environmental exposures and patient-related factors. Furthermore, Intraductal Papillary Mucinous Neoplasm (IPMN), a benign, borderline, low-grade dysplasia arising from pancreatic ductal epithelium, has been reported to be related to PDAC development [9]. Nowadays, the number of IPMNs has significantly increased, but little is known about the specific patterns of cysts’ evolution over time, and not all morphological changes have been detected through imaging modalities [10,11,12]. IPMNs have an estimated prevalence of about 5% in the general population, and recent studies report the IPMN frequency to be much higher among patients with hereditary genetic predisposition syndromes and FPC. Investigating IPMNs’ progression in such high-risk patients is necessary to develop evidence-based management guidelines [13]. Globally, in the whole population, *gBRCA* PVs are estimated to have an incidence rate of 1/300 and 1/800. However, the frequency fluctuates quite a lot, depending on the population and certain ethnic groups harbouring founder mutations, increasing the proportion of *BRCA* PVs in these subgroups. The strongest example of the *BRCA* founder effect belongs to the Ashkenazi Jewish (AJ) population, followed by Dutch, Norwegian and French Canadian ones with founder *BRCA* mutations and therefore an increased risk [14]. Overall, some studies reported a significant heterogeneity in the incidence of PVs and LPVs across different populations [15]. In Italy, little is known about *BRCA1*/2 PV/LPV carriers distributed across the country, and to date, only few studies describe their correlation to a geographical area or to a restricted group [16,17,18,19,20]. Our study aims to present the prevalence and geographic distribution of an inherited *BRCA2* PV identified in HBOC families, followed by our Cancer Genetic Counselling Service and coming from a specific limited area of the eastern coast of Emilia Romagna. In the selected population, we observed 56 carriers of a recurrent and pathogenic *BRCA2* variant, highlighting a specific territorial prevalence. Since within families, some subjects were affected by BC, we confirmed the correlation between its occurrence and the detected variant. Moreover, we estimated the association of the *BRCA2* PV with the risk of developing PDAC, since some of the subjects presenting a family history of HBOC developed PDAC or were observed to exhibit IPNM.

## 2. Materials and Methods

### 2.1. Patients and Samples

Individuals with a suspected HBOC family history were subjected to a first evaluation at Genetic Counselling of the Medical Oncology Division of Rimini Hospital (Rimini, Italy), and, according to the criteria defined by regional protocols, subjects at high risk were invited to undergo *BRCA1/2* genetic testing and to participate in a personalized surveillance program. In particular, over a period ranging from 2012 to 2022, 1012 individuals, including HBOC patients, their relatives and subjects with a cancer family history who met the established inclusion criteria, were recruited and enrolled for genetic counseling. The recruitment was fulfilled according to the Emilia Romagna Regional Program for hereditary risk of breast and ovarian cancer, based on the age at BC/OC onset, number of tumor cases in I- and II-degree relatives, and pathological characteristics of cancers. Genetic testing was performed on all individuals > 18 years old, selected according to the regional program guidelines and Italian Association of Medical Oncology [21,22]. All subjects underwent genetic counseling before testing and after the test result. The probands were affected by BC, OC, PC and Prostate Cancer (PrC). After observing a genetic variant common to some individuals, out of all 1012 subjects, 13 families were considered for the study cohort, and 83 genetic tests were performed both on patients carrying the *BRCA2* PV and on relatives at risk. The present study follows the Declaration of Helsinki and was approved by the institutional review board (Ethics Committee IRST IRCCS-AVR, 2207/2012) after collecting written informed consent from all subjects.

### 2.2. Blood Collection and DNA Extraction

Peripheral blood samples were collected and stored at −80 °C at the Biosciences Laboratory of the IRCCS Istituto Romagnolo per lo Studio dei Tumori “Dino Amadori”. Genomic DNA was extracted using the QIAamp DNA mini Kit (Qiagen, Hilden, Germany) and Maxwell RSC Whole Blood DNA Kit (Promega, Italy) according to the manufacturer’s instructions. DNA was quantified through the Qubit dsDNA BR Assay Kit (Thermo Fisher Scientific, Waltham, MA, USA) using a Qubit fluorometer (Thermo Fisher Scientific, Waltham, MA, USA).

### 2.3. Next-Generation Sequencing

Two panels containing targeted gene *BRCA1/2* were used for next-generation sequencing (NGS). Genetic analysis of the probands and of a part of their relatives was performed through the enrichment protocol TruSight Cancer (Illumina, San Diego, CA, USA), as previously reported [23]. A further part of the subjects was analyzed using the v1.1 SOPHiA Hereditary Cancer Solution (HCS) enrichment protocol (SOPHiA GENETICS, Saint-Sulpice, Switzerland) and output files (FASTQ) uploaded on the SOPHiA DDM Platform v5.5.0 (SOPHiA GENETICS, Saint-Sulpice, Switzerland) for the analysis, as already described by our group [24] Sequencing was performed using the MiSeq sequencer platform (Illumina) and MiSeq Reagent Kit v2 or MiSeq Reagent Kit v3 600 cycles. According to the International Agency for Research on Cancer (IARC) recommendations [25], *BRCA1/2* variants were classified as pathogenic (PV; class 5) and likely pathogenic (LPV; class 4) through the major variants databases and tool prediction software [26].

### 2.4. Statistycal Analysis

Overall data were summarized by frequencies and percentages. The standardized incidence ratio (SIR) with 95% CI was calculated as an estimate of the risk of pancreatic and breast cancers in *BRCA2*-mutated subjects, based on the incidence in the general population of the Rimini province between 2013 and 2017. Person-years at risk were calculated from the date of the genetic test until the date of a cancer diagnosis or 31 December 2021, whichever came first, and were categorized according to the age at diagnosis and sex.

## 3. Results

### BRCA2 Pathogenic Variant and Patient Characteristics

All probands, after appropriate genetic counseling, were genetically tested for germline deleterious variants in *BRCA1/2* genes. Out of 1012 individuals subjected to the genetic test, 103 were observed to be carriers of PV and LPVs in *BRCA1* and 173 in *BRCA2*. Within the population harbouring genetic alterations in *BRCA2* gene, we identified a recurrent germline PV in 13 non-related families, the c.7561del, p.(Ile2521SerfsTer3), a frameshift mutation due to a nucleotide deletion in exon 15, which causes a shift in the reading of the coding sequence, encoding a truncated non-functional protein. Details regarding the *BRCA2*-detected variant are described in Table 1. Since the variant observed in the probands was pathogenic, a genetic test was extended to high-risk family members, leading to the investigation of the variant on a total of 83 subjects. In more detail, our investigation showed that 56 (67.5%) out of 83 individuals were *BRCA2* PV carriers and that 27 (32.5%) were *BRCA2*-negative. Among 56 positive cases, 36 subjects were affected by breast, ovarian, prostate or pancreatic cancer (64.3%), and the remaining part (35.7%) showed no evidence of cancer until our analysis. In particular, out of 36 affected carriers, the diagnoses were of 27 BCs at a mean age of 47 y.o, among which 2 were male BCs, 9 were PDACs at a mean age of 61 y.o., 5 were diagnosed with IPMN, 4 with OCs and 2 with PrCs. Among the 27 affected BC patients, 9 experienced a second BC, in particular 3 ipsilaterally (IL) and 6 contralaterally (CL); thus, out of 36 BC diagnoses, 28 were infiltrating ductal BC (77.9%), 2 were infiltrating lobular (5.5%), 4 were in situ ductal carcinoma (DCIS) (11.1%), and 2 were in situ lobular carcinoma (5.5%). Within some families, some patients experienced more than one cancer, mainly cases of associated BCs and PDACs. For some cases, it was not possible to perform the test, both because it was performed elsewhere and because some subjects had died, but family history allowed us to establish the obligate-carrier state. The clinical history and results of genetic tests carried out on probands and family members were summarized in Appendix A. An illustrative pedigree of a family coming from the geographical area described in our study is shown in Figure 1. From the cohort under analysis, 21 subjects were available for SIR calculation. The age and sex SIR for BC was 24.3, 95% CI 5.0–70.9, while for PC it was 81.5, 95% CI 9.9–294.3, as described in Table 2.

## 4. Discussion

It is well established that PVs/LPVs in *BRCA1/2* genes are associated with an increased risk of HBOCs. Evidences have demonstrated that deleterious *BRCA1/2* variants contribute to the development of other types of cancers, including pancreatic, stomach and male BC [27]. PDAC is still an aggressive disease, with a 5-year overall survival rate of less than 10%. Early diagnosis and consequent surgical resection is the only chance to survive; however, approximately 20% of patients with a localized tumor are susceptible to surgery, and chemotherapy remains the only available regimen option in metastatic PDAC [28]. Most PDAC cases are sporadic; however, about 5–10% show a hereditary predisposition. Several genetic syndromes have been correlated to FPC development, including HBOC, Lynch syndrome (LS), Familial atypical multiple mole melanoma (FAMMM), Familial adenomatous polyposis (FAP), Li–Fraumeni syndrome (LFS), Peutz–Jeghers syndrome (PJS) and Hereditary pancreatitis (HP) [29]. The lifetime risk of PDAC is about 1% for *BRCA1* and 4.9% for *BRCA2* PV/LPVs carriers [30], and studies by the Breast Cancer Linkage Consortium reported a 2.3-fold and 3.5-fold increased risk of PDAC in carriers of *BRCA1-* and *BRCA2*-altered variants, respectively [14]. In addition, a few studies analyzed the prevalence and clinical features of pancreatic cystic lesions in a *BRCA*-tested population in order to establish the risk of pancreatic cysts and cyst-associated cancer, which could have important implications in the context of increasingly accessible genetic testing [31,32,33], but there were significant limitations. The use of next-generation-sequencing (NGS)-based technologies is becoming a new gold standard for determining molecular predictive biomarkers and reliable data on the detection of multiple pathogenic variants, allowing for the screening of thousands of affected individuals [34]. The discovery of new detected variants found in HBOC patients is extremely helpful in aiding clinicians to follow some patients and relatives carrying the same mutations but at a high risk of developing other types of cancer, mostly PDACs. It is worth highlighting that there have been substantial developments in the last years concerning the recognition of *BRCA1/2* PVs, their correlation to PDAC and the targeted therapy implications, especially for the recent FDA approval for Olaparib in the setting of *BRCA*-related PDAC, but unfortunately it remains a malignancy with a very unfavourable prognosis. However, studies are emerging about raising awareness of *BRCA* and about new methods to effectively detect different phenotypes in PDAC, resulting in a subsequent impact on clinical practice and surveillance regimens [35]. Our study aims to obtain a better understanding of a *BRCA*-related gPV in HBOC/PDAC patients and their families. Here, starting from the anamnesis of HBOC cases, we describe the family histories of *BRCA*-related malignancies, which might contribute to the further enrichment of PDAC patients harbouring pathogenic germline variants. There are not many studies reporting the presence of *BRCA2* variants associated with a specific geographical area in Italy and correlated with PDAC. In our cohort, we observed a recurrent pathogenic variant, c.7561delA, p.(Ile2521SerfsTer3), in *BRCA2* gene, only described in HBOC-related studies. Curiously, this variant was found to be restricted to the Rimini district, a limited area of the Emilia Romagna region, and, so far, the observed rate of variants in Italian HBOC families was rather controversial, ranging from 8% to 37%, according to different reports [36]. It is interesting to note that the described variant represents 32% of all *BRCA2* gene variants identified in subjects with suspected HBOC syndrome in the province of Rimini. Among c.7561delA carriers, we observed that 48.2% were affected by BC and, even more importantly, that 16.1% were affected by PDAC, with a percentage that could even increase if it were possible to include additional genetic tests for further familial cases. The frequency and the peculiar geographical distribution of this variant suggest that it may be a founder mutation, as reported both in other populations and ethnic groups in Europe and in the United States, as well as in different areas of Italy, as mentioned above. Furthermore, our study provides evidence of an increased risk of both PDAC and BC in *BRCA2* carriers, although there are limitations due to the small sample size, allowing us to hypothesize that a major prevalence of this germline *BRCA2* PV on the Emilia Romagna eastern coast could be a population-specific genetic feature of positive carriers and could have helpful therapeutic implications for patients and cancer risk prevention in their relatives. In addition, within certain families, some cases of IPMN have been diagnosed. Of course, there are few cases where one could state that they could be related to the variant identified and it was not possible to calculate a statistical correlation for the risk of developing PDAC in these subjects, but it is still important to take them into consideration for any future cases. Our observation gives an overview of the prevalence of a *BRCA2* variant identified in a limited geographic area of Italy, useful for the clinical management of patients. The detection of PVs could have important clinical implications for risk assessment in the family members of mutated patients, especially when taking into consideration the fact that PDAC is spreading in some limited areas and its percentage is increasing. Last but not least, together with the need for a better understanding of the molecular subtypes of *BRCA* PV carriers, the study of pathogenic variants is mandatory for the proper genetic counseling of relatives in order to personalize treatments, increasing the chances of an early detection of BC and PDAC [37].

## 5. Conclusions

Our results show the prevalence of the germline *BRCA2* pathogenic variant c.7561delA, p.(Ile2521SerfsTer3) among individuals recruited in the Rimini district, Italy, consistent with being a high-risk population for HBOC syndrome and for PDAC, for which we calculated the risk of occurrence. This study warrants further targeted cancer prevention programs to properly evaluate high risk individuals.

## Figures and Tables

**Figure 1 cancers-15-02132-f001:**
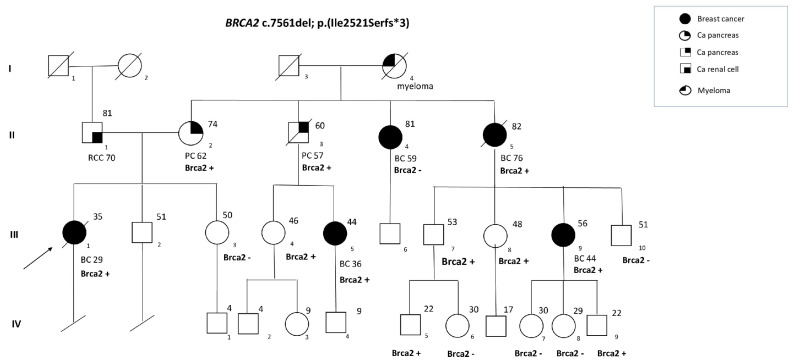
Representative pedigree of a family with *BRCA2* c.7561del variant. I–IV represent generations. Circles represent females, squares represent males. Symbols that are fully dark represent breast-cancer patients. Symbols with a quarter represent subjects with other types of cancer. Symbols with a slash indicate deceased individuals. The arrow points to the proband. BC, breast cancer; PC, Pancreatic Cancer; RCC, Renal Cell Carcinoma; BRCA2 +, presence of PV in BRCA2; BRCA2 −, absence of PV in BRCA2.

**Table 1 cancers-15-02132-t001:** BRCA2 germline pathogenic variant identified in patients and families. IARC, International Agency for Research of Cancer; dbSNP/ClinVar, variant classification according to the Single Nucleotide Polymorphism database (dbSNP) and Clinical Variant (ClinVar).

Gene	Genome Position	c.DNA Change	Protein Change	Variant Type	Consequence	Exon Rank	dbSNP	IARC Class	dbSNP/ClinVar
*BRCA2*, NM_000059.3	13q13.1	c.7561del	p.(Ile2521SerfsTer3)	Deletion	Frameshift	15	rs886040717	C5	Pathogenic

**Table 2 cancers-15-02132-t002:** Age and sex Standardized Incidence Ratio (SIR) of pancreatic and breast cancers in BRCA2-mutated subjects.

Cancer Type	Observed Cases No.	Expected Cases No.	SIR (95% CI)
BC	3	0.12	24.3 (5.0–70.9)
PC	2	0.02	81.5 (9.9–294.3)

## Data Availability

The data presented in this study are available on request from the corresponding author. The data are not publicly available because they are part of genetic data obtained from analyses of patients.

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
