# Peer review of "Prevalence of a BRCA2 Pathogenic Variant in Hereditary-Breast-and-Ovarian-Cancer-Syndrome Families with Increased Risk of Pancreatic Cancer in a Restricted Italian Area"

_cancers, 2023, doi:10.3390/cancers15072132_

Round 1

Reviewer 1 Report

In this manuscript, the authors explored germline BRCA1/2 mutations in a confined region in Italy. The study is helpful for understanding the prevalence of genetic predispositions of cancers. I have the following minor questions

Please state clearly in the result section that 83 individuals from 13 families were subjected to BRCA1/BRCA2 genetic sequencing

Too much usage of abbreviations (such as PC instead of Pancreatic Cancer) makes it difficult to read, please reduce unnecessary aberrations. Please also replace HBOC in the manuscript title

Are all the 56 BRCA pathogenetic variant carriers having the same gremline variant c.7561del? It is confusing in the current manuscript. Also is there any BRCA2 pathogenetic or likely pathogenetic mutations in the cohort?

Fig1 is confusing to read. The authors indicated that symbols with a quarter represent cancer patients. It seems that the individuals with breast cancers were fully dark, while individuals with other cancer types were with other cancer types, and the location of the quarters individual different cancer types. The authors should at least indicate these in figure legends. Otherwise it is difficult to understand. Also the dark colors and quarters appear redundant with the labels that also indicate cancer types.

Reviewer 2 Report

The authors present data on genetically testing a region in Italy for BRCA1/2 likely pathogenic and pathogenic variants.  They attempt to associate this with the prevalence of pancreatic cancer and it's precursor lesion, IPMN.  There may be an association between the genetic and patient/family pancreatic findings, however, the manuscript does not clearly lay out how the authors came to this conclusion.

In it's current form, the authors describe genetically testing a cohort of patients using genetic testing criteria.  Beyond that description, it does not provide further explanation on how they came to the families presented in supplement table 1.  Did they only test 13 families, and found only one variant in the BRCA2 gene?  That seems highly unusual, but if that is the case, then it should be highlighted further.  Then if that is the only variant that was ascertained after assessing 1012 families, that needs to emphasized.  Then further expanding on those BRCA2 positive families and their pancreatic phenotype should be laid out.  It does seem unusual that 9/13 BRCA2 positive families have a pancreatic phenotype leading do some sort of founder effect of this variant.

Minor comments:

BRCA1 and BRCA2 should be italicized throughout, and a common format used (e.g. BRCA1/2 vs BRCA)

BRCA2 wild type should be changed to BRCA2 negative or BRCA2 true negative for the familial cases.  No one is truly BRCA wild type.

Round 2

Reviewer 2 Report

the authors have addressed my comments